

# Whole gut microbiome composition of damselfish and cardinalfish before and after reef settlement

Darren J. Parris[1], Rohan M. Brooker[2], Michael A. Morgan[1], Danielle L. Dixson[2] and Frank J. Stewart[1]

[1] School of Biology, Georgia Institute of Technology, Atlanta, GA, United States
[2] School of Marine Science and Policy, University of Delaware, Newark, DE, United States

Corresponding author
Frank J. Stewart,
frank.stewart@biology.gatech.edu

## ABSTRACT

The Pomacentridae (damselfish) and Apogonidae (cardinalfish) are among the most common fish families on coral reefs and in the aquarium trade. Members of both families undergo a pelagic larvae phase prior to settlement on the reef, where adults play key roles in benthic habitat structuring and trophic interactions. Fish-associated microbial communities (microbiomes) significantly influence fish health and ecology, yet little is known of how microbiomes change with life stage. We quantified the taxonomic (16S rRNA gene) composition of whole gut microbiomes from ten species of damselfish and two species of cardinalfish from Lizard Island, Australia, focusing specifically on comparisons between pelagic larvae prior to settlement on the reef versus post-settlement juvenile and adult individuals. On average, microbiome phylogenetic diversity increased from pre- to post-settlement, and was unrelated to the microbial composition in the surrounding water column. However, this trend varied among species, suggesting stochasticity in fish microbiome assembly. Pre-settlement fish were enriched with bacteria of the Endozoicomonaceae, Shewanellaceae, and Fusobacteriaceae, whereas settled fish harbored higher abundances of Vibrionaceae and Pasteurellaceae. Several individual operational taxonomic units, including ones related to *Vibrio harveyi*, *Shewanella sp.*, and uncultured *Endozoicomonas* bacteria, were shared between both pre and post-settlement stages and may be of central importance in the intestinal niche across development. Richness of the core microbiome shared among pre-settlement fish was comparable to that of settled individuals, suggesting that changes in diversity with adulthood are due to the acquisition or loss of host-specific microbes. These results identify a key transition in microbiome structure across host life stage, suggesting changes in the functional contribution of microbiomes over development in two ecologically dominant reef fish families.

## INTRODUCTION

Animals harbor diverse microbial communities that influence key aspects of host health, development, and behavior. In most vertebrates, the majority of microorganisms inhabit the gastrointestinal (GI) tract at an abundance of potentially trillions of cells whose

collective genome may be orders of magnitude larger than that of the host (*Whitman, Coleman & Wiebe, 1998*; *Backhed et al., 2005*; *Gill et al., 2006*). A wealth of studies in both fish and mammalian systems now confirm the importance of a gut microbiome to host health, fitness, and development (*Rawls, Samuel & Gordon, 2004*; *Bates et al., 2006*; *Chung et al., 2012*; *Lee & Hase, 2014*), with alterations of microbiome composition affecting such properties as host immunity, susceptibility to toxins, adiposity, efficiency of nutrient acquisition, behavior and mood, and chemical signaling among conspecifics (*Turnbaugh et al., 2006*; *Round & Mazmanian , 2009*; *Heijtz et al., 2011*; *Greenblum, Turnbaugh & Borenstein, 2012*; *Theis et al., 2013b*; *Zheng et al., 2013*; *Wada-Katsumata et al., 2015*).

Despite recognition of the important role of microbiomes in host ecology (*Wong & Rawls, 2012*), microbiomes remain unexplored for the vast majority of ecologically relevant taxa, including the most diverse of the vertebrate groups, the teleost fishes. Fish account for almost half of all vertebrate species on earth and span a wide spectrum of habitats, host ranges, physiologies, and ecological strategies. Studies of fish gut microbiomes have focused primarily on commercial or model species, with most targeting only a small number of host or microbial taxa (*Roeselers et al., 2011*; *Clements et al., 2014*; *Llewellyn et al., 2014*). These studies raise the possibility of a core set of fish gut microbes shared across diverse hosts, notably a dominance by bacteria of the Gammaproteobacteria and Firmicutes, including unique fish-associated strains of the Vibrionales and Clostridia (*Spanggaard et al., 2000*; *Al-Harbi & Naim Uddin, 2004*; *Martin-Antonio et al., 2007*; *Sullam et al., 2012*; *Xing et al., 2013*; *Llewellyn et al., 2014*) Significant variation in the fish gut microbiome has also been reported with changes in taxonomic composition shown to affect host immunity, nutrient acquisition, and epithelial differentiation (*Rawls, Samuel & Gordon, 2004*; *Bates et al., 2006*; *Bates et al., 2007*; *Cheesman & Guillemin, 2007*; *Cheesman et al., 2010*; *Kanther & Rawls, 2010*; *Ghanbari, Kneifel & Domig, 2015*). Fish microbiome composition has been linked to diverse factors including host type (*Ye et al., 2014*; *Givens et al., 2015*; *Hennersdorf et al., 2016*), trophic ecology and diet (*Bolnick et al., 2014a*; *Bolnick et al., 2014b*; *Miyake, Ngugi & Stingl, 2015*; *Sullam et al., 2015*), and environmental conditions (e.g., salinity; *Sullam et al., 2012*; *Schmidt et al., 2015*)

The composition of the fish microbiome can also change significantly over an individual's lifespan (*Bakke et al., 2015*; *Burns et al., 2015*; *Llewellyn et al., 2015*). However, the timing, determinants, and magnitude of these shifts are unclear for most taxa, and likely highly variable among species with differing life histories. On coral reefs, for example, >90% of fish species undergo a dispersing or planktonic stage as either larvae or juveniles, followed by an adult phase marked by settlement into a constrained territory on the reef (*Leis, 1991*). For such species, microbiome structure is hypothesized to be shaped by a combination of factors that co-associate with host life stage, including diet shifts, immune system and GI tract development, and differences in the composition of the external microbial community between pre- and post-settlement habitats.

Comparing reef fish microbiomes over developmental stages will help identify shifts in the contribution of the microbiome to host health and behavior, thereby helping reach a holistic understanding of reef ecology. Collectively, coral reefs harbor over 2,500 fish species engaged in a complex web of interactions including predation, herbivory,

corallivory, and symbiosis (*Bellwood et al., 2004*; *Allen, 2014*). These interactions together modulate material and energy transport and ecosystem structuring on coral reefs, with important consequences for reef-building corals that rely on fish waste for nutrients or fish herbivory to limit competition with benthic macroalgae (*Mumby et al., 2006*; *Burkepile & Hay, 2008*; *McCauley et al., 2010*). Such interactions may also be important vectors for moving microbes between reef habitats. A handful of studies have characterized gut microbiomes of coral reef fish, identifying a role for both diet type and host identity in shaping microbiome composition (*Smriga, Sandin & Azam, 2010*; *Miyake, Ngugi & Stingl, 2015*). However, the microbiomes of the vast majority of reef fishes, including some of the most abundant and ecologically relevant taxa, remain uncharacterized.

Damselfish (Pomacentridae) and cardinalfish (Apogonidae) are among the most diverse and abundant families of reef fishes and exhibit similar ecological strategies. Species within these families occupy critical positions in reef food webs as generalist planktivorous consumers, subsisting on a variety of algae, plankton, and benthic invertebrates (*Emery, 1973*; *Emery & Thresher, 1980*; *Marnane & Bellwood, 2002*). These fish also constitute major fractions of the diet of larger piscivorous reef fish (*Beukers-Stewart & Jones, 2004*). Most species of damselfish and cardinalfish spend their first 12–35 days of life as members of the off-reef plankton community before settling back onto the reef as juveniles (*Wellington & Victor, 1989*; *Leis, 1991*; *Leis, Sweatman & Reader, 1996*). Post-settlement, damselfishes (including members of the *Chromis*, *Pomacentrus*, and *Dascyllus* genera) occupy distinct microhabitats using various types of coral for habitat and cover (*Wilson et al., 2008*). Many Pomacentrids are highly territorial and influence turf algae composition through grazing (*Hinds & Ballantine, 1987*; *Klumpp & Polunin, 1989*). Apogonids also occupy coral microhabitats during the day when they are mostly inactive but leave coral cover at night to feed throughout the water column (*Gardiner & Jones, 2005*). Fish species of the Apogonidae often dominate nocturnal planktivore assemblages (*Marnane & Bellwood, 2002*). In this study we quantified microbiome taxonomic composition (16S rRNA gene diversity) in whole guts from replicate individuals from ten species of damselfish and two species of cardinalfish (Table S1). We focused on comparisons between pre-settlement planktonic larvae and post-settlement individuals (juveniles through to adults) to determine whether microbiome composition varied between ontogenetic stages and if so whether this variation was related to differences in the microbial composition of the surrounding water column.

## MATERIALS AND METHODS

### Sample collection

Fishes were collected over a 3–4 day period from the fringing reefs surrounding Lizard Island, Australia (14°40.08′S 145°27.34′E) in February 2014. Water depths at collection sites were approximately 5–10 m. Larval pre-settlement individuals (8–16 mm total length, Table S1) were defined as samples collected from floating light traps deployed overnight at a single site. These traps sampled the upper meter of the water column above the reef. Post-settlement stage juveniles/adults (16–36 mm total length, Table S1) were individuals hand-collected on SCUBA at depth from nearby reefs using dip nets. Post-settlement fish

in this study were likely sexually immature (juveniles) and recently settled given their small size (sexual maturity in most Pomacentrids typically occurs at lengths >60 mm, *Schmale, Hensley & Udey, 1986*; *Kavanagh, 2000*). Net capture of settled fish followed mild sedation with a clove oil/ethanol mixture dispensed via spray bottle. All fish were euthanized by immersion in a seawater-ice slurry combined with an overdose of clove oil and stored in ethanol for further analysis. Water column samples ($n = 11$) were collected on SCUBA at the same depth and location of post-settlement fish collection sites and the light trap site using sterile plastic 1L bottles. Bottles were first opened within the water column to ensure collection at the desired location and prevent contamination. Microbial biomass in the sampled water was concentrated onto 0.2 µm Polyvinylidene Flouride (PVDF) filters through filtration. Filters were immediately placed in ethanol in cryovials and stored at room temperature until processing. Fish were identified to the lowest possible taxonomic unit, typically either genus or species, using Indo-Pacific reef fish field guides based on morphology. In total, 49 pre-settlement individuals were sampled (Table S1), including individuals of the damselfish *Pomacentrus moluccensis* ($n = 4$), *P. chrysurus* ($n = 4$), *P. nagasakiensis* ($n = 4$), *P. amboinensis* ($n = 4$), *P. wardii* ($n = 4$), *P. bankanensis* ($n = 4$), *P. coelestis* ($n = 4$), *Dascyllus aruanus* ($n = 3$) and an unidentified *Chromis* species ($n = 7$). Two species of pre-settlement stage cardinalfish, *Ostorhinchus doederleini* ($n = 4$) and an unidentified *Apogon* species ($n = 7$), were also collected. A total of 24 post-settlement individuals were sampled (Table S1), including individuals of 5 species of damselfish (*P. moluccensis* ($n = 6$), *P. chrysurus* ($n = 3$), *A. polyacanthus* ($n = 3$), *Dascyllus aruanus* ($n = 5$) and an unidentified *Chromis* sp. ($n = 3$)) and 1 cardinalfish (*O. doederleini* ($n = 4$)). Note that the individuals of the unidentified *Chromis* species (collected both pre and post-settlement) were almost certainly either *Chromis viridis* or *Chromis atripectoralis*, both of which are similar ecologically and closely related. All research was reviewed and conducted under the guidelines of Georgia Tech IACUC #A14063 and Great Barrier Reef Marine Park Authority collection permits G13.36166.1 and animal ethics permit A1920. These methods were approved by Georgia Tech IACUC and Great Barrier Reef Marine Park Authority for use in this study.

## DNA extractions

Whole fish samples were removed from storage containers and rinsed thoroughly with fresh, filter-sterilized ethanol to remove (potentially) surface attached microbes. As dissection of only the intestinal contents was not possible due to the small size of most pre-settlement individuals, whole gut contents (stomach + intestines) were excised by dissection with a sterile razor from each individual. Excised contents were placed in a sterile 1.5 ml centrifuge tube and frozen until extraction. Bulk DNA was extracted from gut contents using the Qiagen DNeasy blood and tissue kit according to manufacturer instructions. To process water samples, each collection filter was transferred from ethanol into a clean, 1.5 ml centrifuge tube. The ethanol remaining in the original storage tube was vacuumfiltered onto a new 0.2 µm PVDF filter, which was then dried at room temperature and pooled with the initial collection filter. DNA was extracted from filters using a phenol:chloroform method. Briefly, cells were lysed by adding lysozyme (2 mg in 40 µl of lysis buffer per sample)

directly to the pooled filters, capping, and incubating for 45 min at 37 °C. Proteinase K (1 mg in 100 µl lysis buffer, with 100 µl 20% SDS) was added and samples incubated for an additional 2 h at 55 °C. The lysate was removed, and DNA was extracted once with phenol:chloroform:isoamyl alcohol (25:24:1) and once with chloroform:isoamyl alcohol (24:1). Extracted DNA was concentrated by spin dialysis using Ultra-4 (100 kDa, Amicon) centrifugal filters and stored frozen.

## PCR amplification and sequencing

High-throughput sequencing of dual-indexed PCR amplicons spanning the V3-V4 hypervariable regions of the 16S rRNA gene was used to assess gut microbiome taxonomic composition. Amplicons were synthesized using Platinum® PCR SuperMix (Life Technologies) with primers F515 (5'-GTGCCAGCMGCCGCGGTAA-3') and R806 (5'-GGACTACHVGGGTWTCTAAT-3', *Caporaso et al., 2011*), with both primers modified to include sample-specific barcodes and Illumina sequencing adapters according to *Kozich et al. (2013)*. Ten nanograms of starting DNA was used as template for each PCR reaction. Amplification was performed using denaturation at 94 °C (3 min), followed by 30 cycles of denaturation at 94 °C (45 sec), primer annealing at 55 °C (45 sec), primer extension at 72 °C (90 sec), and a final extension at 72 °C for 10 min. Amplicon products were verified using gel electrophoresis, purified using Diffinity RapidTip2 PCR purification tips (Diffinity Genomics, NY), and quantitated fluorometrically using the Qubit (Life Technologies). Amplicons from different samples were pooled at equimolar concentrations and sequenced on an Illumina MiSeq using a 500 cycle kit with 30% PhiX added to increase sequence diversity. All raw sequences are available in the NCBI Sequence Read Archive under BioProject ID PRJNA290348.

## Sequence analysis

Barcoded sequences were de-multiplexed and trimmed (length cutoff: 100 bp) and filtered to remove low quality reads (cutoff: Phred score of 25, averaged over all bases) using Trim Galore (http://www.bioinformatics.babraham.ac.uk/projects/trim_galore/). Paired-end reads were then merged using FLASH (*Magoč & Salzberg, 2011*), with an average read length threshold of 250 bp, and a fragment length threshold of 300 bp. Merged reads were analyzed using the QIIME pipeline (*Caporaso et al., 2010*). In QIIME, chimeric sequences were identified using USEARCH (*Edgar, 2010*) and removed from the input dataset. Merged non-chimeric sequences were clustered into Operational Taxonomic Units (OTUs) at 97% sequence similarity using open-reference picking with the UCLUST algorithm (*Edgar, 2010*). Singleton OTUs (those represented by only one sequence read in the combined dataset) were also removed at this step. Taxonomy was assigned to each OTU by comparison with the Greengenes database 13_5 release (*DeSantis et al., 2006*). Cyanobacterial and chloroplast sequences were manually removed from the OTU tables as these sequences can be assumed to reflect non-host-associated taxa, potentially those brought into the gut through food passage. Selected OTUs were queried via BLASTN against the NCBI-nr database to identify closest relatives.

OTU counts were rarefied (10 iterations) to 4,262 sequences per sample and the rarified OTU table was used in downstream analysis. A Bonferroni-corrected ANOVA was used

to identify microbial families significantly enriched in pre- vs. post-settlement individuals with family composition data expressed as percentage of total reads. Alpha and beta diversity statistics for pre- and post-settlement individuals were calculated in QIIME using the core_diversity_analyes.py command. The unweighted Unifrac metric was used to conduct principal coordinates analysis to visualize beta diversity. Significant differences in beta diversity were identified using a two-sided $t$-test (Bonferroni-corrected). Core microbiome analysis was performed in QIIME using the compute_core_microbiome.py script to identify individual OTUs shared by at least 70% of all individuals per sample grouping (all pre-settlement individuals vs. all post-settlement individuals). This script was also used to estimate the average number of OTUs shared between random pairs of individuals from each life stage, based on 100 random iterations per life stage. Indicator analysis was performed with the indicspecies package in R to identify genera (indicator taxa) significantly enriched according to life stage, taking into consideration both the extent to which an OTU is exclusive to a life stage (A component), as well as the relative frequency at which the OTU occurs in all individuals within a stage (B component)(*Dufrene & Legendre, 1997*; *Fortunato et al., 2013*). Only those taxa occurring in more than half of all samples (*B* component > 0.5) and for which permutation testing yielded *p*-values <0.05 are included in the output. Indicator values range from 0 to 1 with higher values reflecting stronger relationships between indicator taxa and the tested sample groupings (pre- vs. post-settlement).

## RESULTS AND DISCUSSION

### General microbiome characteristics

A total of 49 pre-settlement and 24 post-settlement individuals representing ten Pomacentridae species and two Apogonidae species were collected from Lizard Island, Australia. Of these, five species—the pomacentrids *Pomacentrus moluccensis*, *P. chrysurus*, *Dascillus aruanus*, and an unidentified *Chromis* sp. (either *C. viridis* or *C. atripectoralis*) and the apogonid *Ostorhinchus doederleini*—were represented by both pre- and post-settlement individuals, with the remaining species represented only in the pre-settlement sample set (*n* = 6 species) or only in the post-settlement set (*n* = 1).

A total of 3,503,605 16S rRNA gene sequences were obtained after quality control filtering (average: 41,709 per sample; range: 4262–238,858). After rarefaction, OTU counts per sample were highly variable, from 44 to 557 OTUs, with averages of 163 (±59 standard deviation (SD)) and 245 (±102 SD) for pre- and post-settlement stages, respectively. As observed in other fish species (*Spanggaard et al., 2000*; *Al-Harbi & Naim Uddin, 2004*; *Martin-Antonio et al., 2007*; *Sullam et al., 2012*; *Xing et al., 2013*), Pomacentrid and Apogonid gut microbiomes were dominated by Gammaproteobacteria of the Pseudoaltermonadaceae, Endozoicimonaceae, Vibrionaceae, and Shewanellaceae with these Families constituting 80% (±10.3 SD) and 67% (±6.7 SD) of all sequence reads in pre- and post-settlement fishes, respectively (Fig. S1), although key differences in the relative abundance of microbial families were also evident (discussed below, Table 1).

It is important to note that our data do not differentiate communities based on location in the GI tract, as the small size of individuals prohibited separation of intestine

**Table 1** Median % abundance of bacterial Families[*] in pre- versus post-settlement damselfish and cardinalfish (pooled).

| Family | Pre | | Post | | |
| --- | --- | --- | --- | --- | --- |
| | Med | MAD | Med | MAD | Fold |
| Vibrionaceae | 18.12 | 14.20 | 41.91 | 28.69 | 2.31 |
| Endozoicimonaceae[**] | 11.02 | 9.50 | 0.14 | 0.14 | 76.40 |
| Pseudoalteromonadaceae | 1.28 | 1.15 | 0.45 | 0.42 | 2.83 |
| Rhodobacteraceae | 1.24 | 1.01 | 1.35 | 1.22 | 1.09 |
| Flavobacteriaceae | 1.07 | 0.89 | 0.61 | 0.60 | 1.75 |
| Alteromonadaceae | 0.71 | 0.63 | 0.85 | 0.75 | 1.20 |
| Shewanellaceae | 0.68 | 0.65 | 0.11 | 0.11 | 5.99 |
| Oceanospirillaceae | 0.56 | 0.51 | 0.15 | 0.15 | 3.74 |
| Pseudomonadaceae | 0.29 | 0.24 | 0.49 | 0.45 | 1.70 |
| Moraxellaceae | 0.26 | 0.23 | 0.02 | 0.02 | 10.31 |
| Pirellulaceae[**] | 0.01 | 0.01 | 0.83 | 0.81 | 81.97 |
| Chromatiales, Unknown | 0.10 | 0.08 | 0.21 | 0.20 | 2.03 |

**Notes.**

[*]only includes Families with median abundance >0.2% in either pre or post datasets.

[**]significant change pre vs. post ($P < 0.05$, ANOVA, Bonferroni corrected) MAD, median absolute deviation. Fold, fold increase in median (pre to post); underlined indicates post to pre.

from stomach contents. Thus, we cannot distinguish between resident gut microbiome members, which might be relatively more common in the intestine, versus members derived from food contents or passively from seawater intake, which might be relatively enriched in the stomach. Both host-adapted as well as transient microbiome members may nonetheless exert effects on host health, as either symbionts or potentially as inocula for resident populations. However, gut microbiome composition (weighted Unifrac; Pomacentridae + Apogonidae combined) of both pre- and post-settlement stages differed significantly from that of the surrounding seawater ($p = 0.028$; Fig. 1), which was instead enriched in Gammaproteobacteria of the Halomonadaceae family, Alphaproteobacteria, and Bacteroidetes (Fig. S2). Moreover, microbiome composition did not differ significantly between water collection sites (Fig. S3), suggesting fine-scale variation in local environments alone should not influence microbiome composition and an overall minimal effect of source water on gut microbiome structure.

## Microbiome diversity and composition

Microbiomes showed a general trend toward greater complexity following settlement, although this trend was variable among species. The phylogenetic composition of the combined (Pomacentrid + Apogonid) microbiomes changed significantly from pre- to post-settlement groups (weighted Unifrac metric; $p = 0.028$), with pre-settlement microbiomes showing less intra-individual variation compared to those of post-settlement fish (Fig. 1) and overall lower phylogenetic diversity (PD) (average: $20.3 \pm 7$ SD, versus $28.7 \pm 12$ SD post-settlement; $p = 0.001$, $t$-stat $= 3.5$; Fig. 2). Both pre- and post-settlement microbiomes exhibited substantially lower PD than the surrounding water column community (PD: $79.4 \pm 23.5$ SD; $p = 0.001$). Of the five species for which both

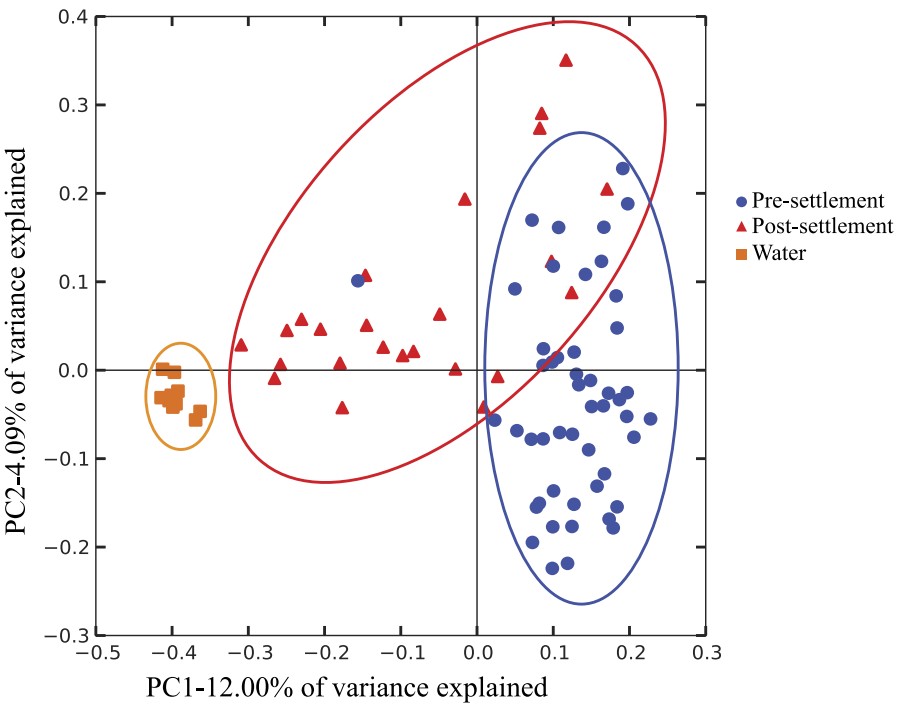

**Figure 1 Compositional relatedness of microbiome samples.** Principal coordinate analysis based on unweighted Unifrac distance with sequence data rarified to 4,262 sequences per sample (49 pre-settlement, 24 post-settlement, 11 seawater samples). According to a two-sided $t$-test (Bonferroni-corrected), pre-settlement and post-settlement individuals are significantly different from one another ($p < 0.05$) and both are significantly different from the water column ($p < 0.05$).

pre- and post-settlement samples were available, three - *Chromis sp., O. doederleini,* and *D. aruanus* - exhibited significantly higher PD post-settlement (Fig. 3, $p = 0.001$). Greater microbiome diversity with age has been observed in other fish species, with suggested linkages to increases in diet complexity (*Bolnick et al., 2014a*; *Bolnick et al., 2014b*; *Miyake, Ngugi & Stingl, 2015*; *Sullam et al., 2015*) or potentially to differentiation of the GI tract into distinct niches with development. In contrast, elevated microbiome diversity in early life was observed in *P. chrysurus* and *P. moluccensis* (Fig. 3, $p = 0.001$), consistent with data from lab-reared zebrafish (*Stephens et al., 2016*). This pattern suggests that pre-settlement microbiomes in these species are influenced primarily by transient associations with bacteria from the exterior environment or by higher diet complexity at the planktonic stage. As an individual matures, its microbiome may become more specialized in response to diverse factors, including diet shifts, immune system development, physiological and chemical changes along the GI tract, or decreased connectivity with the external environment. Indeed, species-specific microhabitat usage (i.e., selection of distinct coral types by different species) has been observed in some of the hosts examined here, including *P. moluccensis* and *P. nagasakiensis*. However, all settled hosts in our dataset are assumed to be generalist planktivores with fairly similar post-settlement diets based on prior work (*Emery, 1973*; *Emery & Thresher, 1980*; *Marnane & Bellwood, 2002*). Although this study does not allow us to quantify the influence of each of the factors discussed above on developmental shifts

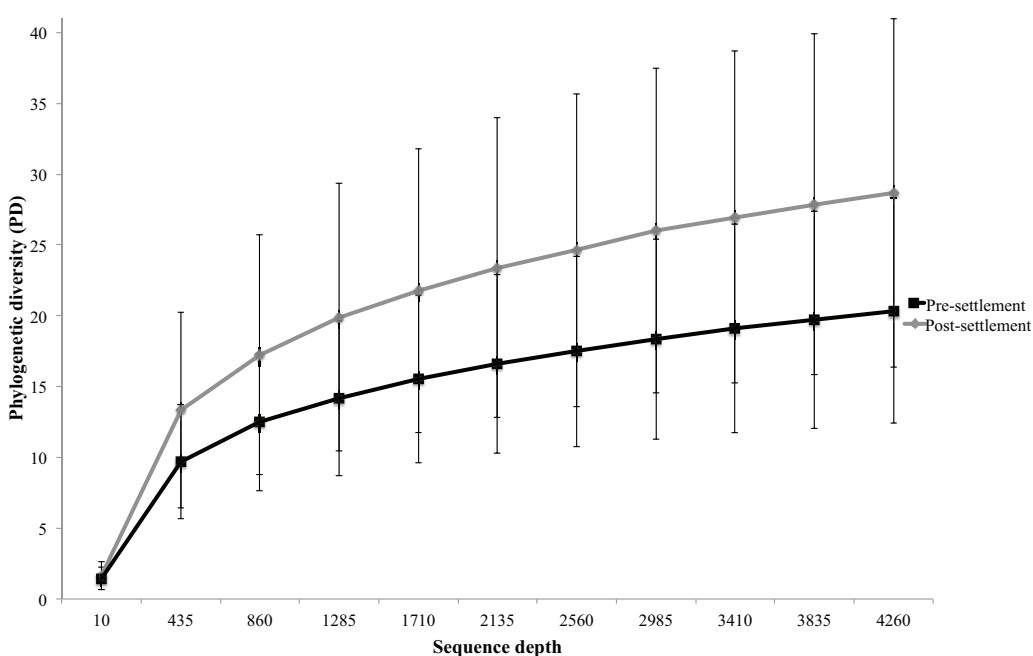

**Figure 2 Microbiome phylogenetic diversity (PD) as a function of sequence depth in pre- and post-settlement fish (all species combined, based on data rarified to 4262 sequences per sample).** Error bars are ± 1 standard error.

in microbiome composition, our data, combined with results from prior studies, suggest that microbiome diversity is dynamic over life stages and assembly processes vary on a per-species basis.

Key microbial taxa differed in abundance between pre- and post-settlement microbiomes (Table 1), although microbiome composition was relatively consistent within hosts per age group (Fig. S3). At the microbial family level, pre-settlement microbiomes were significantly enriched (>70-fold) in bacteria of the Endozoicimonaceae, while settled fish harbored over 2-fold higher median abundances of the Vibrionaceae and an 80-fold enrichment of planctomycetes of the Pirellulaceae (Table 1). Differences in the relative abundance of microbes at the family level did not appear to be driven by a single host. For example, all pre-settlement individuals were enriched in bacteria of the Endozoicomonaceae compared to post-settlement hosts (Fig. S4). At the microbial genus level, both life stages were marked by sets of indicator taxa (Table 2). Indicator taxa represent individual OTUs showing significant enrichment (as a percent of total sequence reads) in one life stage vs. the other according to permutation testing ($p < 0.05$, *Dufrene & Legendre, 1997 Fortunato et al., 2013*). The number of indicators was considerably higher for settled fish (38) compared to larvae (5; Table 2), consistent with the increase in diversity associated with post-settlement, perhaps reflective of an overall expansion in microbial niche breadth as discussed above. The marine bacterial genera *Kordia* (Flavobacteriia) and *Halomonas* (Gammaproteobacteria, Oceanospirillales) were the strongest indicators of pre-settlement stage (Table 2), followed by members of the *Arcobacter* (Epsilonproteobacteria, Campylobacterales), *Oceanospirillum* (Gammaproteobacteria, Oceanospirillales), and

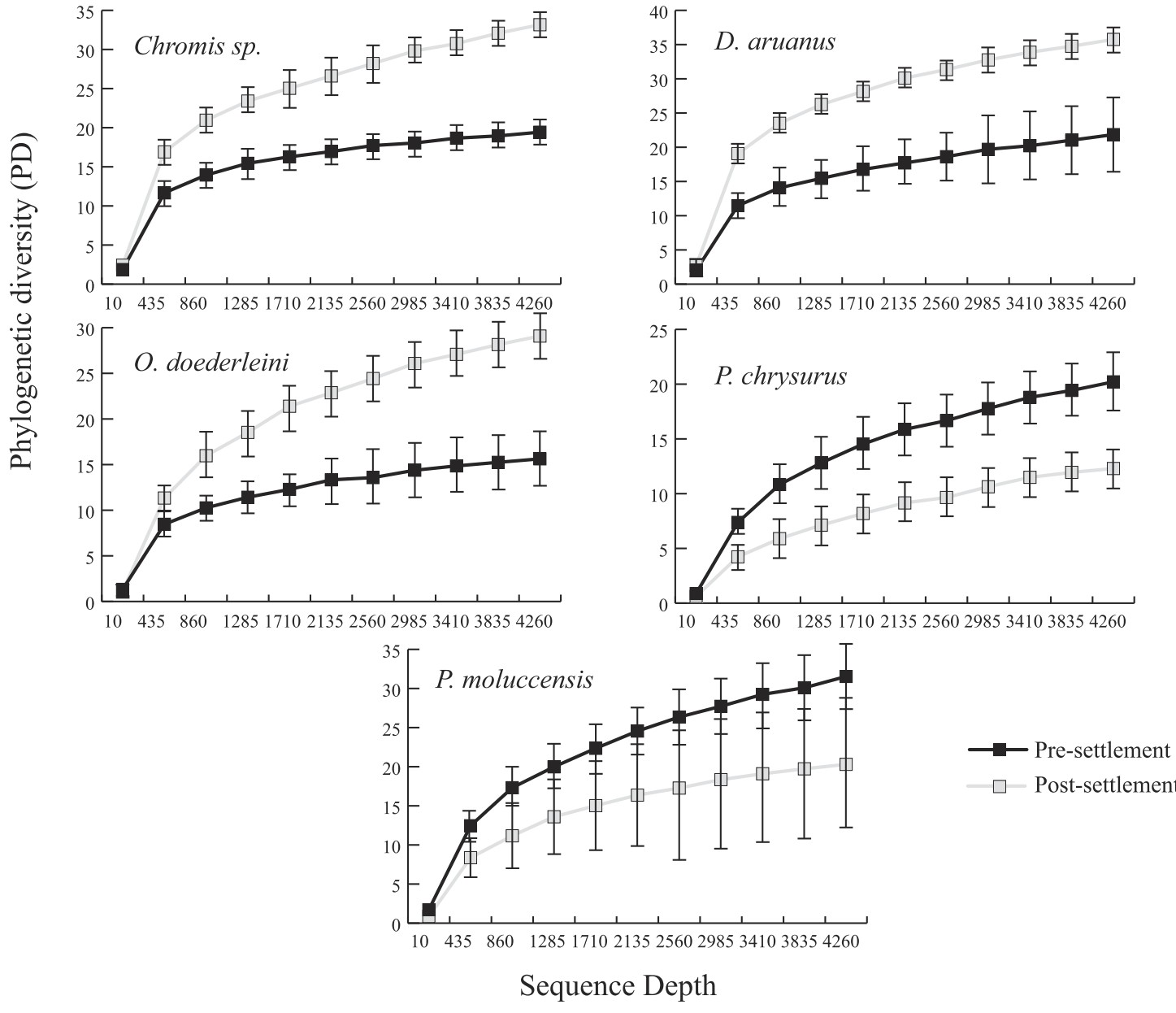

**Figure 3** **Microbiome phylogenetic diversity (PD) as a function of sequence depth in five species represented in both pre- and post-settlement datasets.** Error bars are ±1 standard error.

*Idiomarina* (Gammaproteobacteria, Alteromonadales), all taxa common to the marine environment but not often observed in fish microbiomes. Pre-settlement enrichment implicates these taxa as important during early development, more successful at colonizing and growing in young fishes, or commonly associated with pre-settlement food sources. Indicators of post-settlement spanned a wide phylogenetic breadth, with several classifiable only to the order or family level, and the strongest indicators including diverse members of the Gammaproteobacteria (Vibrionaceae, Portiera), Firmicutes (Epulopiscium), Alphaproteobacteria (Kiloniellales, Hyphomicrobiaceae),

**Table 2  Indicator genera associated with pre- and post-settlement damselfish and cardinalfish (pooled).**

| Indicator genera | Indicator for pre or post settlement fish | Indicator value | *p*-value |
|---|---|---|---|
| Kordia | Pre | 0.943 | 0.001 |
| Halomonas | Pre | 0.896 | 0.001 |
| Arcobacter | Pre | 0.869 | 0.001 |
| Oceanospirillum | Pre | 0.823 | 0.001 |
| Idiomarina | Pre | 0.751 | 0.002 |
| unclassified Vibrionaceae | Post | 0.997 | 0.01 |
| unclassified Pirellulaceae | Post | 0.922 | 0.001 |
| unclassified Kiloniellales | Post | 0.904 | 0.001 |
| unclassified Gammaproteobacteria | Post | 0.9 | 0.001 |
| Coraliomargarita | Post | 0.863 | 0.001 |
| Portiera | Post | 0.857 | 0.001 |
| Epulopiscium | Post | 0.857 | 0.001 |
| unclassified Hyphomicrobiaceae | Post | 0.851 | 0.001 |
| Verrucomicrobium | Post | 0.816 | 0.001 |
| unclassified Rhizobiales | Post | 0.786 | 0.001 |
| Rhodospirillaceae | Post | 0.779 | 0.001 |
| unclassified Acidimicrobiales OCS155 | Post | 0.778 | 0.001 |
| unclassified Oceanospirillales | Post | 0.771 | 0.001 |
| unclassified Peptostreptococcaceae | Post | 0.771 | 0.001 |
| unclassified Myxococcales | Post | 0.768 | 0.001 |
| unclassified Altermonadales OM60 | Post | 0.763 | 0.013 |
| unclassified Phyllobacteriaceae | Post | 0.761 | 0.001 |
| Ferrimonas | Post | 0.75 | 0.001 |
| unclassified Thiohalorhabdales | Post | 0.746 | 0.001 |
| unclassified Flavobacteriales | Post | 0.743 | 0.001 |
| unclassified Cryomorphaceae | Post | 0.733 | 0.038 |
| unclassified Altermonadales | Post | 0.733 | 0.038 |
| unclassified Myxococcales OM27 | Post | 0.728 | 0.001 |
| Balneola | Post | 0.724 | 0.001 |
| Crocinitomix | Post | 0.723 | 0.001 |
| unclassified Flammeovirgaceae | Post | 0.722 | 0.002 |
| unclassified Altermonadaceae | Post | 0.722 | 0.035 |
| unclassified Francisellaceae | Post | 0.712 | 0.001 |
| Saprospria | Post | 0.692 | 0.003 |
| Clostridium | Post | 0.692 | 0.016 |
| unclassified Rickettsiales | Post | 0.688 | 0.001 |
| Turicibacter | Post | 0.682 | 0.001 |
| unclassified Deltaproteobacteria GMD14H09 | Post | 0.678 | 0.006 |
| unclassified Pasteurellaceae | Post | 0.676 | 0.002 |
| unclassified Actinomycetales | Post | 0.675 | 0.001 |
| Altermonadaceae HTCC2207 | Post | 0.67 | 0.001 |
| unclassified Comamonadaceae | Post | 0.664 | 0.004 |
| Flavobacterium | Post | 0.656 | 0.003 |

Verrucomicrobia (Coraliomargarita), and Planctomycetes (Pirellulaceae) (Table 2). Enrichment of these taxa in fully developed, settled fish suggests the host GI tract as a specific niche for members of these groups.

## Comparison of the core microbiome between pre and post-settlement individuals

Although egg-tending behavior is common in many pomacentrids, fish are presumed sterile at birth and the fish gut microbiome is assumed to be inoculated primarily from the environment and via food items early in life (*Nayak, 2010*). If the pre-settlement, pelagic diet is similar across species, it is reasonable to predict that larval individuals of different fish species will share a greater number of microbial taxa (i.e., larger core microbiome). The core microbiome might then be expected to shrink post-settlement as species adapt to specific diets and microhabitats. We explored this prediction in two ways. First, we determined the average number of OTUs shared between randomly sampled pairs of pre-settlement individuals compared to post-settlement individuals. Contrary to our expectation, the average count of shared OTUs did not vary with life stage (22.8 vs 23.2 for pre- and post-settlement, respectively). Second, we compared the richness and composition of the core microbiome between life stages, following normalization to a uniform sample size (# of host individuals) per stage. Consistent with a recent analysis of microbiomes from 15 coastal fish species (*Givens et al., 2015*), no single microbial OTU was present in all individuals of either group (pre or post) in our study. Contrary to our prediction, the richness of the core microbiome was similar for both pre and post-settlement stages; for example, 17 OTUs were shared across 70% of pre-settlement individuals, compared to 15 for post-settlement (Fig. 4). Furthermore, many of the same OTUs were present in the core microbiomes identified at each stage. For example, of the OTUs occurring in 70% of total microbiome samples (pre- and post combined), 12 (of 16 total) were present in both the pre- and post-settlement core sets (70% threshold) when evaluated separately (Fig. 4), indicating conservation of a subset of key microbial members across settlement. Together, these results suggest processes affecting assembly and diversity of the core microbiome, when defined at the OTU level, are relatively constant from pre- to post-settlement.

Many of these core OTUs were absent from or at much lower abundances in seawater samples (with the exception of 3 OTUs), indicating selective accumulation in the fish gut (Fig. 4). Notably a single OTU with high sequence similarity to *Vibrio harveyi* (Vibrionaceae, Genbank accession KX380754.1) comprised an average of 18%–20% of the total sequence reads in both pre- and post-settlement fish, but only 2% of the seawater community (Fig. 4). *Vibrios* are often described as opportunistic pathogens (*Karunasagar et al., 1994*; *Austin & Zhang, 2006*). For example, metagenome sequences related to *Vibrio* species, including *V. harveyi*, from the guts of farmed adult turbot were enriched in genes encoding potential virulence functions (*Xing et al., 2013*). However, the wide distribution of *V. harveyi* in healthy marine hosts, observed here and in other studies of diverse invertebrates and fish (*Onarheim et al., 1994*; *Guerrero-Ferreira et al., 2013*), suggests that the primary niche of this bacterium may be commensal or even mutualistic with the host, potentially with a role in protein degradation and digestion (*Ray, Ghosh & Ringø, 2012*). Interestingly, the
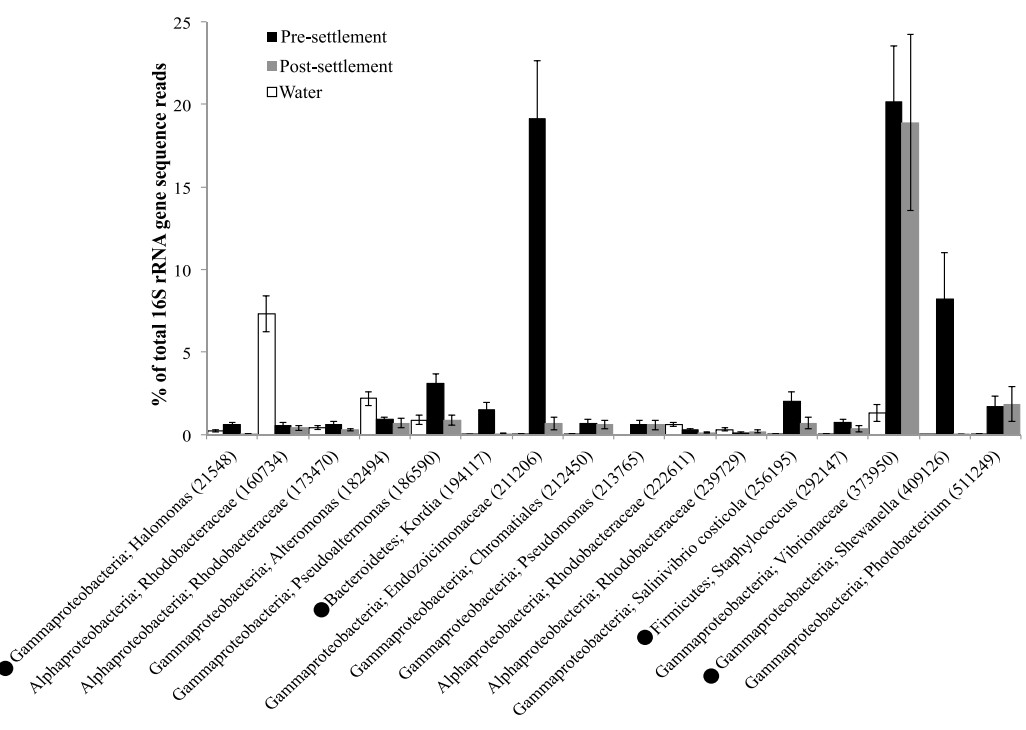

**Figure 4** **Taxonomic identity and relative abundance of OTUs detected in greater than 70% of all fish microbiome samples (pre-settlement ($n = 49$) and post-settlement ($n = 24$) samples) combined for analysis.** ● indicates that an OTU was shared among 70% of the pre-settlement samples when this sample set was evaluated independently, but was not detected in >70% of samples in the post-settlement set when evaluated independently. The relative abundances of these OTUs in the water column samples ($n = 11$) are included for comparison. Taxonomic classifications are at the level of genus whenever possible. Error bars are ±1 standard error.

Vibrionaceae family as a whole (multiple OTUs) was over twice as abundant in post-settlement fish (Table 1) and therefore may be more typical of mature gut microbiome communities.

The proportional abundance of other core microbiome taxa varied significantly between life stages. The 16 taxa detected in 70% of all microbiome samples, a core set dominated by OTUs belonging to the Gammaproteobacteria (Fig. 4), constituted ∼60% of the total gut microbiome sequence reads in pre-settlement fish but only ∼25% in settled fish, indicating that these OTUs may be important early in life. Notably, OTUs closely related to *Shewanella putrafaciens* (Genbank accession KP967510.1) and to an uncultured bacterium of the family Endozoicomonaceae (Genbank accession LN626318.1) constituted 8% and 18% of the pre-settlement microbiome, respectively, but less than 1% of the post-settlement community (Fig. 4). To our knowledge, this is the first report of Endozoicomonaceae occupying a large proportion of the fish gut microbiome. Bacteria of this family, within the widely distributed Oceanospiralles group, exhibit diverse heterotrophic metabolisms (*Neave et al., 2014*) and are commonly identified as symbionts in marine invertebrates (*Bayer et al., 2013a*; *Bayer et al., 2013b*; *Forget & Kim Juniper, 2013*; *Nishijima et al., 2013*; *Beinart et al., 2014*;

*Hyun et al., 2014*), suggesting invertebrate prey as a potential vector of transport into the fish microbiome. The functional properties and potential for long-term residency of these bacteria in reef fishes remain to be verified. In contrast, Gammaproteobacteria of the genus *Shewanella*, have been detected widely in fish. Indeed, *Shewanella putrefaciens*, the species most closely related (97%) to the core OTU in this study, has been used as a probiotic to increase growth and pathogen resistance in juvenile sole (*Lobo et al., 2014*).

Other members of the core OTU set spanned diverse bacterial divisions, but occurred at lower abundances (Fig. 4). These included members of genera containing known fish pathogens, such as *Photobacterium* (*Chabrillón et al., 2005*), *Pseudoalteromonas* (*Pujalte Domarco et al., 2004*), *Pseudomonas* (*Wakabayashi et al., 1996*), and *Halomonas* (*Austin, 2005*). However, these groups also contain potentially beneficial members, as certain *Pseudoalteromonas* and *Alteromonas* strains can inhibit bacterial pathogens in fish (*Gatesoupe, 1999*) and *Photobacterium* species are known bioluminescent symbionts in diverse fish hosts (*Ruby & Nealson, 1976*; *Ast et al., 2007*). Core OTUs also included a Chromatiales-affiliated taxon with 100% similarity to the uncultured gill symbiont of *Ifremeria nautilei* (*Beinart et al., 2015*; Genbank accession KF780855.1), a member of the *Kordia* genus resembling a heterotrophic isolate from the gut of marine polychaetes (*Choi et al., 2011*, Genbank accession NR_117471.1), and *Salinivibrio costicola*, a halotolerant, facultatively anaerobic bacterium first isolated from a hypersaline pond (*Huang et al., 2000*). Three OTUS matching marine Rhodobacteraceae, a broadly distributed aquatic group that has also been detected in reef surgeonfish (*Miyake, Ngugi & Stingl, 2015*), were also prevalent across samples. However, two of these OTUs were more abundant in water samples, suggesting these taxa may be transient members of the gut community. Together, these data identify a relatively species-poor (e.g., 16 OTUs at 70%) but abundant (35%–65% of total 16S sequences) core microbiome across the damselfish and cardinalfish sample set, including potentially pathogenic or beneficial bacteria, and a subset of core OTUs (12 of 16) that persist across pre- and post-settlement life stages. Persistence over development may be an important indicator of commensal or mutualistic taxa ubiquitous to reef fish.

## CONCLUSION

These results describe a diverse gut microbiome in two abundant reef fish families and highlight the importance of life stage in structuring microbiome composition. A trend toward greater microbiome diversity in settled (older) individuals was observed in the pooled dataset, potentially explained by the hypothesis that pre-settlement fish of diverse species are more similar to each other physiologically compared to adults, occupy the same niche (pelagic zone), and acquire gut microbes from a common environmental pool, perhaps due to similarities in diet. These factors could homogenize the pre-settlement microbiome across diverse host species, with microbiomes diversifying after fish settle on the reef and transition to adult feeding roles. However, this pattern is not uniform across host species, suggesting a need for discretion in concluding general trends in microbiome succession over development. Our data also identify core microbes common to pre- and post-settlement fish of several host species. Persistence of these microbes

over major life transitions, regardless of variable trends in bulk microbiome complexity, may implicate these microbes as particularly important to host health and physiology. Differences in microbiome assembly between pre and post-settlement fish are likely driven by a combination of factors, including physiological changes associated with development, changes in diet and potentially feeding frequency, and varying connectivity with microbes from the external environment. More research is needed to disentangle the relative contributions of these determinants over the settlement transition.

## ACKNOWLEDGEMENTS

We thank the staff of Lizard Island Research Station for logistical support.

### Funding

This work was enabled by support from the Simons Foundation (Grant 346253 to FJS). The funders had no role in study design, data collection and analysis, decision to publish, or preparation of the manuscript.

### Grant Disclosures

The following grant information was disclosed by the authors:
Simons Foundation: 346253.

### Competing Interests

The authors declare there are no competing interests.

### Author Contributions

- Darren J. Parris conceived and designed the experiments, performed the experiments, analyzed the data, wrote the paper, prepared figures and/or tables, reviewed drafts of the paper.
- Rohan M. Brooker conceived and designed the experiments, performed the experiments, reviewed drafts of the paper.
- Michael A. Morgan performed the experiments.
- Danielle L. Dixson conceived and designed the experiments, performed the experiments, contributed reagents/materials/analysis tools, reviewed drafts of the paper.
- Frank J. Stewart conceived and designed the experiments, analyzed the data, contributed reagents/materials/analysis tools, wrote the paper, prepared figures and/or tables, reviewed drafts of the paper.

### Animal Ethics

The following information was supplied relating to ethical approvals (i.e., approving body and any reference numbers):

All research was reviewed and conducted under the guidelines of Georgia Tech IACUC #A14063 and Great Barrier Reef Marine Park Authority collection permits G13.36166.1 and animal ethics permits A1920. These methods were approved by Georgia Tech IACUC and Great Barrier Reef Marine Park Authority for use in this study.

## Data Availability

All raw data (sequences) are available in the NCBI Sequence Read Archive (GenBank) under BioProject ID PRJNA290348.

## Supplemental Information

Supplemental information for this article can be found online at http://dx.doi.org/10.7717/peerj.2412#supplemental-information.

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
