# Peer review of "Whole gut microbiome composition of damselfish and cardinalfish before and after reef settlement"

_PeerJ, doi:10.7717/peerj.2412_

## Round 0.1 · original submission · Major Revisions

The reviewers provided very detailed reviews and raised important points that will help improve your manuscript.

Reviewer 1 ·

Basic reporting

This article is within the scope of the journal; the format and coherence of the submission are well done. Appropriate figures are included, and described clearly.

The introduction gives good background covering both the interest in and knowledge gaps for the microbial ecology field, as well as a clear statement of how the work aims to address this gap. Given the subject matter, I also recommend the authors also read and include an additional reference: Llewellyn, M. S., McGinnity, P., Dionne, M., Letourneau, J., Thonier, F., Carvalho, G. R., … Derome, N. (2016). The biogeography of the atlantic salmon (Salmo salar) gut microbiome. The ISME Journal, 10(5), 1280–1284.

The authors are encouraged to include more information about the host animals with respect to relevant ecology and biology.

Experimental design

This work is clearly primary research, with the raw data already available publically, thank you. Ethical standards for vertebrates are met.
Generally the methods are quite clear and reproducible, however I have a few questions and requests for the authors:
1. How is pre-settlement determined vs post? Is there morphology shift, length definitions? If Length, please include lengths of fishes in table S1. Additionally, if maturation might have major effect, juvenile vs. adult differentiation is needed.
2. As great value is placed in the use of data like these in future studies, please include the sequences of the primers used for 16S amplification in your methods. While the primers are named and cited, this particular pair may be misinterpreted by others due to changes the the widely used Earth Microbiome Project. This is of particular importance to marine environments which popular primers for this region are known to be quite distinct in their taxonomic coverage (e.g. capture or exclusion of the SAR11 clade).

The stated knowledge gap is well identified and real. The research seeks to uncover additional factors over which microbial composition may be expected to change for teleost fish. Specifically this work examines whether microbial composition varies or not between ontogenetic stages (pre-settlement larvae and post-settlement juveniles & adults). Life-stage is a major factor to consider when investigating host-associated microbial communities, and is rarely examined for any animal.

While the sampling and processing seems well considered, the experimental setup itself may need increased depth of sampling for multiple individuals within test groups. Additionally, there is little treatment of the confounding factors often linked to species differences. For example, it would be better to have some treatment of diet (via stomach contents) or citation of previous rigorous study would be appropriate, as diet is known to be a major factor.

Validity of the findings

The work presented here uses 16S rRNA profiling to examine shifts in gut-microbiome composition across ontogenetic life-stages. It is my opinion that while the analyses are reasonably done, the design does not support all of the conclusions put forth by the authors.

There is little treatment of the distinguishing features of lifestyle between species, which may have large bearing on the hypotheses that adult fish might have less diverse gut communities, particularly as settled damselfish may be largely herbivorous (as noted by the authors). Additionally, contrary to the stated assertion that there is little parental care in fishes, the species studied here are known to have brood-tending behaviors, which may indeed affect the microbial communities associated from first spawn to hatching. Overall, there is some question as to whether pre- and post- settlement animals collected at the same time are comparable, particularly as seasonally-linked “seeding” of the microbiome may happen.

Additionally, please address the following two corrections:
Line 289: Epulopiscium is a famous member of the Firmicutes
Figure 4: OTU 292147 - is this correctly notated as "not detected in post-settlement"? Appears otherwise here.

Comments for the author

While it is understandably difficult to control for all variable of the system, it would be good for the authors to address the potential effects of the host ecology and physiology more fully.

This being said, your contribution to broadening the scope of host microbiome studies across more taxa and more factors than diet is appreciated and will clearly add to our understanding.

·

Basic reporting

Submission meets PeerJ policies. The study cites studies that examine the fish gut microbiome but fails to cite/mention other key studies which have looked at fish diversity and respective gut community diversity (Wong et al. 201; Hennersdort et a.l 2016; Ghanbari et al. 2015 and see papers cited in Sullam et al. 2012 and Givens et al. 2015).

Experimental design

Paper presents research comparing the gut microbiome of damselfish and cardinalfish pre- and post-settlement and includes clearly defined research questions and objectives. Methods are clearly written, but additional information needs to be added regarding sequencing reaction (read length, paired end reads?), whether sequences have deposited in a publicly-available database (i.e. NCBI SRA or otherwise), and the statistical methods for Table 1 data. Additionally, authors need to specify whether they looked at recovery rates from their gut samples. The authors frequently report on the % contribution of certain bacteria within the gut, but need to clarify whether this % represents contributions of OTUs or takes into account the relative abundance (# of total sequence reads) for these reported OTUs. The uncertainty in the author's meaning makes the Results section unclear and authors need to clarify exactly what data they are using for statistical analyses and are presenting and discussing in the Results and Discussion section.

Validity of the findings

Authors need to clarify several points in Results and Discussion section prior to publication. The authors focus on cardinalfish and damselfish, but use a number of different species within these groupings. One major concern is that the authors do not use all the same species pre- and post-settlement and fail to adequately present data and/or address intra-species (among fish) and inter-species (between species) variability. Did specific species have core microbiomes different from those presented for the whole group? Instead the authors combine all pre- and post-settlement results which potentially glosses any differences due to specific host or the particular area (habitat) where fish were collected. Likewise, authors do not discuss variability in the seawater microbiome collected at different times or locations -- how did the seawater and fish collected at the same time/place compare to overall combined results of all pre- and post-settlement fish and seawater environment. Furthermore, authors present on "indicator taxa" but do not define this term or explain how indicators were defined for their dataset. Considering, "indicator taxa" result is a substantial component of the Results, this needs to be detailed in the Methods section and clarified in the Results and Discussion section.

Due to the size of the pre-settlement gut tract, authors used the entire gut defined as the stomach and gut tract. Were the stomachs emptied or diet examined? If not, authors are likely including the bacteria-associated with or in the diet in addition to what was part of the actual fish's gut microbiome. Authors need to clarify this point and mention how this influences the results and conclusions from those results.

Comments for the author

See comments above regarding Experimental Design and Validity of Findings.

On Line 45, please specify if you are referring to general gut microbiome studies or fish gut microbiome studies. There have been numerous studies looking at the human gut and health and physiology but less with other species.

Line 128 and elsewhere -- no need to capitalize the word "genus" or "family." You capitalize the family name, i.e. Vibrionaceae.

Lines 376-383, this lengthy sentence is difficult to follow and needs to be revised for clarity.

When presenting % ranges, include mean and standard deviation.

Provide NCBI accession numbers for "similar" organisms referenced in Results and Discussion section.

Figure 1, clarify whether groupings are significant and by what analysis.

---

## Round 0.2 · Minor Revisions

Corrections brought by the authors were appropriate regarding the comments made by the two reviewers. However, many of the new references were not correctly cited. I also noted two minor corrections to be made in the new text lines. I suggest these corrections being done before the paper is accepted for publication.

Lines 53–54: Should be “Wong and Rawls” or the proper reference should be added.

Lines 101–102: Should be “Beukers-Stewart and Jones” or the proper reference should be added.

Line 111: Reference to Gardiner and Jones has been added in the text, but not in the Reference section.

Line 128: I think the interval should be 14–36 mm based on data presented in Table S1. Am I right?

Lines 265–268: The sentence is long and somewhat confusing. I would suggest splitting text in two different sentences (the first one could end after “should not influence microbiome composition”, and the second one could read, “The microbiome composition data also suggests minimal effect of source water…”).

Lines 295–296: The two references to Emery et al. do not fit the References listed page 23. Please bring appropriate corrections.

---

## Round 0.3 · accepted · Accept

Thank you for these last quick corrections.